# Using Function space theory for understanding intermediate layers

**Shai Dekel**
Tel-Aviv University and WIX AI
shaidekel6@gmail.com

**Oren Elisha**
Tel-Aviv University and Microsoft Israel
orenelis@gmail.com

## Abstract

The representational change of input along the intermediate layers is an important aspect of understanding deep learning architectures. To this end, we propose an approach that relies on the foundation of Function Space theory. In particular, we argue that a weak-type Besov smoothness index can quantify the geometry of the clustering in the feature space of each layer. Therefore, our approach may provide an additional perspective for understanding data-models fit in the setting of deep learning. While using a different framework and perspective, the experiments we performed are in line with the results described by Tishby & Zaslavsky (2015) and Montavon et al. (2010) in the sense that for well-performing trained networks, the quality of the representation increases from layer to layer. Our approach could also be used for addressing generalization (Zhang et al., 2016), (Kawaguchi et al., 2017) as we also show that the Besov smoothness of the layer representations of the training set decreases as we add more mis-labeling.

## 1 Function space analysis for neural network architectures

### 1.1 Function space approach

A function space is a class of functions bundled with a norm that assigns a non-negative magnitude to every function in the class. In many cases, we are interested in the collection of those functions for which the definition of the norm makes sense and is finite Tao (2008). For example, the functions that have a quantity nature, such as $L_p$ spaces, or some smoothness characteristics such as Sobolev spaces. One of the practical aspects of this field, is finding the functions representation which could provide a correspondence with its Function Space. A well-known example is the representation of functions as Fourier series and the correspondence between the Fourier coefficients and the functions error decay rate. In this position paper, we will be using Geometric Wavelets for representing functions along with Besov space analysis, which is the right mathematical setup for adaptive approximation using wavelets (Dekel & Leviatan, 2005), (DeVore, 1998). As shown in (Elisha & Dekel, 2017), the weak-type Besov smoothness besov indication could describe the geometry of the clustering of the training set in the feature space of each layer.

We begin with an instructive example that could demonstrate our functional perspective for neural network architectures. Assume we are presented with a set of gray-scale images of dimension $\sqrt{n_0} \times \sqrt{n_0}$ with $L$ class labels. Assume further that a deep network has been successfully trained to classify these images with relatively high precision. This allows us to extract the representation of each image in each of the hidden layers. To create a representation at layer 0, we concatenate the $\sqrt{n_0}$ rows of pixel values of each image, to create a vector of dimension $n_0$. We also normalize the pixel values to the range $[0, 1]$. Since we advocate a function-theoretical approach, we transform the class labels into vector-values in the space $R^{L-1}$ by assigning each label to a vertex of a standard simplex. Thus, the images are considered as samples of a function $f_0 : [0, 1]^{n_0} \to R^{L-1}$. In the general case, there is no hope that there exists geometric clustering of the classes in this initial feature space and that $f_0$ has sufficient 'weak-type' smoothness (as illustrated by our experiments below). Thus, a transform into a different feature space is needed. We thus associate with each $k$-th layer of a DL network, a function $f_k : [0, 1]^{n_k} \to R^{L-1}$ where the samples are vectors created by normalizing and concatenating the feature maps computed from each of the images. Interestingly enough, although the series of functions $f_k$ are embedded in different dimensions $n_k$, through the

simple normalizing of the features, our method is able to assign smoothness indices to each layer that are comparable. We claim that for well performing networks, the representations in general 'improve' from layer to layer and that our method captures this phenomena and shows the increase of smoothness.

In some sense this formalism resembles to the 'probe' approach described by Alain & Bengio (2016), which detects the immediate suitability for a linear classifier in each layer. However, in our approach we treat the input representation of each layer as a discrete dataset $\{x_i \in \Omega_0, f(x_i)\}_{i=1,,m}$, in some convex bounded domain $\Omega_0 \subset R^n$, and search for an efficient representation of the underlying function. This representation should overcome the complexity, geometry and possibly non-smooth nature of the values of the underlying function. Such evaluation could be done for each intermediate layer after a simple value normalization of the features and response variables. The function Space approach is trying to revile the sparsity and the geometric properties of this representation rather than its accuracy.

## 1.2 WAVELET DECOMPOSITION OF RANDOM FOREST (RF)

Wavelet decomposition of RF (Elisha & Dekel, 2016) provides a representation of a predictive mode in a formalism that enables smoothness analysis. The RF algorithm (Breiman, 2001) constructs diverse subdivisions of the initial domain $\Omega_0$ into decision trees $\mathcal{T}_j$. At each stage of the subdivision process, the RF forms a partition of any node in a convex domain $\Omega \subset R^n$ by an hyper-plane partition into two convex subdomains $\Omega', \Omega'', \Omega' \cup \Omega'' = \Omega$. For the two children nodes this process also enables an association of two multivariate polynomials $Q_{\Omega'}, Q_{\Omega''} \in \Pi_{r-1}(R^n)$, of fixed (typically low) total degree r − 1. Observe that for any given subdividing hyperplane, such approximating polynomials can be uniquely determined for $p = 2$, by least square minimization. For the simple case $r = 0$, these polynomials are nothing but the average values/labels of the points that belongs to the node.

Denoting by $\mathbf{1}_{\Omega'}$, the indicator function over the child domain $\Omega'$, we use the polynomial approximations $Q_{\Omega'}, Q_{\Omega}$, computed by the local minimization described at (Elisha & Dekel, 2017) and define

$$\psi_{\Omega'}(x) := \psi_{\Omega'}(f)(x) := \mathbf{1}_{\Omega'}(x)(Q_{\Omega'}(x) - Q_{\Omega}(x)), \tag{1}$$

as the **geometric wavelet** associated with the subdomain $\Omega'$ and the function $f$, or the given discrete dataset $\{x_i, f(x_i)\}_{i=1,,m}$.

Each wavelet $\psi_{\Omega'}$, is a 'local difference' component that belongs to the detail space between two levels in the tree, a 'low resolution' level associated with $\Omega$ and a 'high resolution' level associated with $\Omega'$. The norm of a wavelet is computed by

$$\|\psi_{\Omega'}\|_p^p = \int_{\Omega'} (Q_{\Omega'}(x) - Q_{\Omega}(x))^p dx.$$

Using the weights that are assigned to the trees in the RF (e.g. $w_j = 1/J$), we obtain a wavelet representation of the entire RF

$$\tilde{f}(x) = \sum_{j=1}^{J} \sum_{\Omega \in \mathcal{T}_j} w_j \psi_{\Omega}(x). \tag{2}$$

The theory (see Dekel & Leviatan (2005), Elisha & Dekel (2016) ) tells us that sparse approximation is achieved by ordering the wavelet components based on their norms

$$w_{j(\Omega_{k_1})} \|\psi_{\Omega_{k_1}}\|_p \geq w_{j(\Omega_{k_2})} \|\psi_{\Omega_{k_3}}\|_p \geq \cdots \tag{3}$$

Thus, the adaptive M-term approximation of a RF is

$$f_M(x) := \sum_{m=1}^{M} w_{j(\Omega_{k_m})} \psi_{\Omega_{k_m}}(x). \tag{4}$$

with the notation $\Omega \in \mathcal{T}_j \Rightarrow j(\Omega) = j$.

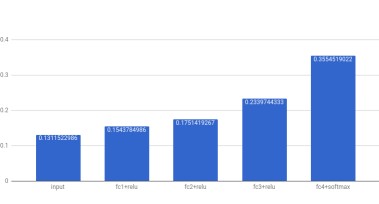 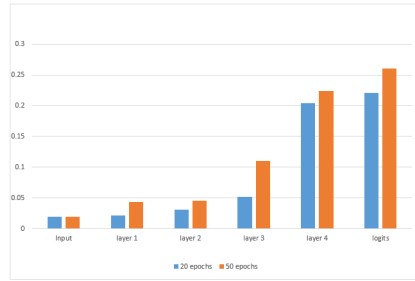

Figure 1: Smoothness analysis of the layer representations of Urban8K' (left) and CIFAR10 (right)

Table 1: Smoothness analysis of mis-labeled image images

| Mis-labeling | 0% | 10% | 20% | 30% | 40% |
|---|---|---|---|---|---|
| MNIST smoothness | 0.28 | 0.106 | 0.084 | 0.052 | 0.03 |
| CIFAR10 smoothness | 0.204 | 0.072 | 0.053 | 0.051 | 0.003 |

### 1.3 REVILING THE WEAK-TYPE SMOOTHNESS OF INTERMEDIATE LAYERS

The formal definition and construction of the Besov space of a function with a RF partitioning $|f|_{\mathcal{B}^{\alpha,r}_\tau(\mathcal{F})}$ could be found at Elisha & Dekel (2016). Approximation Theory supports cases where the function is not even continuous, and describe a correspondence between the sparsity of a function and its Besov smoothness index $\alpha$. This notion of smoothness indicates that the higher the index $\alpha$ for which $|f|_{\mathcal{B}^{\alpha,r}_\tau(\mathcal{F})}$ is finite, the smoother the function is. The theory in Elisha & Dekel (2017) shows that for the case $r = 1$, we can infer the smoothness index $\alpha$ from the error in the wavelets representation

$$\sigma_M(f) := \|f - f_M\|_p \tag{5}$$

without dependence on the dimension of $n_k$. One could numerically model $log(\sigma_M) \sim log(c_k) - \alpha log(M)$, $M = 1, ..., \tilde{M}$, and then find $c_k, \alpha_k$ through least squares. Finally, we set $\alpha_k$ as our estimate for the 'critical' Besov smoothness index of $f_k$. this evaluation could be done solely on the training set for each of the k representation layers.

## 2 APPLICATIONS AND EXPERIMENTAL RESULTS

### 2.1 SMOOTHNESS ANALYSIS ACROSS DEEP LEARNING LAYERS

In the left side of Figure 1 we see how the clustering is 'unfolded' by the network, as the Besov $\alpha$-index increases from layer to layer, using the "Urban8K" audio data [1] at the layers of the DeepListen model [2]. In right side of Figure 1 we see a clear indication of how the smoothness begins to evolve during the training after 20 epochs and the 'unfolding' of the clustering improves from layer to layer. We also see that the smoothness improves after 50 epochs, correlating with the improvement of the accuracy.

### 2.2 SMOOTHNESS ANALYSIS OF MIS-LABELED DATASETS

Following Zhang et al. (2016) and Kawaguchi et al. (2017), we applied random mis-labeling to the MNIST and CIFAR10 image sets at various levels. We randomly picked subsets of size $q\%$ of the size of dataset, with $q = 10\%, 20\%, 30\%, 40\%$, and then for each image in this subset we picked a random label. We then trained the networks on the misclassified datasets. We emphasize that the goal of this experiment is to understand generalization and automatically detect the level of corruption solely from the smoothness analysis of the training data. we created a wavelet decomposition of RF on the representation of the training set at the last inner layer of the network.

---

[1] https://serv.cusp.nyu.edu/projects/urbansounddataset/
[2] https://github.com/jaron/deep-listening

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
