# OpenReview forum: "USING FUNCTION SPACE THEORY FOR UNDERSTANDING INTERMEDIATE LAYERS"
_ICLR.cc/2018/Workshop — Reject_

### Official Review · AnonReviewer3 · 2018-03-04
**Interesting new method, although the article could be clarified, and an important piece of literature is missing.**

**Rating:** 7
**Confidence:** 3

**Review:**

Summary
The authors propose a method to define the "smoothness" of the representation computed by a neural network (or a layer of a neural network). If I understood correctly, the idea is to train a random forest on top of the learned representation. A set of "wavelets" can be associated to this random forest. These wavelets define Besov spaces with different degrees of smoothness, and it is possible to determine the "most regular" Besov space that the classification function belongs to.
The authors test their method with networks trained on Urban8K and CIFAR10. They find that the smoothness is increasing accross layers. They also investigate the link between smoothness and mislabeling of the dataset.

Novelty: the idea of looking at the smoothness of representations is not novel (see "Cons"), but the method to evaluate smoothness is new (as far as I know).
Clarity: can be improved; there are more comments about this in the last part of this review.
Significance: good.
Quality: the article seems rigorous.

Pros
1. This article proposes a notion of smoothness that has a solid mathematical basis. This notion is a priori sufficiently sophisticated to capture subtle properties of neural networks, and maybe allow comparison between different architectures. So, in my opinion, it looks promising.
2. The experiments of the authors suggest a strong link between smoothness of the learned representation and generalization. This part of the work seems a bit preliminary, but it might be very interesting if it was further developed.

Cons
1. The authors do not discuss the implications of their work for the understanding of deep learned representations. Knowing that the representations become smoother and smoother accross the layers is an interesting and "self-sufficient" properties, but the method proposed by the authors tells more than that. It allows to precisely quantify the smoothness; as highlighted by the authors, it is also based on a solid mathematical theory. How do the authors expect these properties to be useful in future work?
2. Studying the smoothness of learned representations is not something new. It has in particular been studied by Oyallon in "Building a regular decision boundary with deep networks", although with less sophisticated tools (nearest neighbors instead of Besov spaces). I think a comparison with this work is necessary.
(There might be other works on the subject, but, since I am not a specialist, I do not know them.)

Typos / minor remarks
- "Function Space" should not have capital letters (as well as "Approximation Theory" in 1.3).
- "mis-labeling" -> "mislabeling"?
- First paragraph: there should be parentheses around "Tao (2008)". What is a "quantity nature"? A verb is also lacking in this sentence. I do not understand "One of the practical ... space." The expression "error decay rate" for a function is unclear. In the line before the last one, the second "besov" should be removed.
- I did not understand the last paragraph of Subsection 1.1.
- The first paragraph of Subsection 1.2 is, in my opinion, a bit unclear. What is a node of a convex domain? What are the polynomials associated to? How can the approximating polynomials be uniquely defined, since they have not been required to satisfy a single property? I recommend rewriting this paragraph in a more pedagogical style.
- Last line of the first paragraph of 1.2: "belongs" should be "belong".
- Subsection 1.2: The notation j(Omega) should be defined before it is used, not afterwards. I also do not think that "J" is formally defined somewhere. By the way, how is J chosen in the numerical tests?
- In Equation (3), k_3 should be k_2.
- In the pictures, the legends are too small, and difficult to read.
- End of 1.1, and title of 1.3: "revile" -> "reveal".
- Subsection 1.3: I do not understand "a correspondence ... smoothness index alpha". A function can be (Besov-)smooth without being sparse. I think the "sparsity" is the sparsity of its wavelet decomposition.
- End of 1.3: what does the index k represent in c_k and alpha_k? Why is log(sigma_M) modeled as log(c_k) - alpha log(M) (with a dependency on k for c_k, but not for alpha)?
- Last sentence of 1.3: capital letter missing.
- My understanding of this article is that the smoothness is computed by defining a random forest on top of a deep representation. Is this correct? If yes, I think it could be explained more clearly in the article.
- Subsection 2.2: why is smoothness computed only for the last layer? Do the other layers behave the same? If no, why? Which network architectures are used? Do they perfectly classify almost all training samples, even with mislabeling?
- Last sentence: capital letter missing.

---

### Official Review · AnonReviewer2 · 2018-03-09
**It is hard to say that this result is new or non-trivial.**

**Rating:** 4
**Confidence:** 3

**Review:**

- Outline -
This paper investigates an effect of an low-rank approximation to tensors from an activation of DNNs.
Authors experimentally show that when tensors is reconstructed as low-rank, accuracy by DNNs is decreased and values of parameters are changed.

- Comment -
In my opinion, it is hard to say that this result is new or non-trivial.
Basically, it is well known that volume of approximation error decreases as a number of ranks increases.
In addition, the approximation effect of tensors for DNNs is already studied.
Thus, the decrease of accuracy is obviously explained by the low-rank approximation.
It is hard to say their finding is new.

---

### Official Review · AnonReviewer1 · 2018-03-12
**Smoothness (in a learned wavelet basis) is claimed to increase with depth and training time of a neural network.**

**Rating:** 3
**Confidence:** 4

**Review:**

The paper relies heavily on recent (cited) results of the authors that define and study the function smoothness results and their associated wavelet basis. It is unclear from this manuscript what usefulness the smoothness measure scalar provides in empirical use of networks. Further, the experimental evaluation is not sufficient to make the above claims on smoothness. For example, "we also see that the smoothness improves after 50 epochs, correlating with the improvement of the accuracy" is true but not enough to deliver on the stated promise that smoothness could be used to predict accuracy.

The writing needs further clarifying - it can be difficult to understand sentences like "The function Space approach is trying to revile the sparsity and the geometric properties of this representation rather than its accuracy."

---

### Decision · Program_Chairs · 2018-03-20
**ICLR 2018 Workshop Acceptance Decision**

**Decision:**

Reject

**Comment:**

Based on the reviews, this paper has not been accepted for presentation at the ICLR workshop. However, the conversation and updates can continue to appear here on OpenReview.